# Clinical and Genotypical Features of False-Negative Patients in 26 Years of Cystic Fibrosis Neonatal Screening in Tuscany, Italy

**DOI:** 10.3390/diagnostics10070446

**Published:** 2020-07-01

**Authors:** Giovanni Taccetti, Matteo Botti, Vito Terlizzi, Maria Chiara Cavicchi, Anna Silvia Neri, Valeria Galici, Gianfranco Mergni, Claudia Centrone, Diego G. Peroni, Filippo Festini

**Affiliations:** 1Tuscany Reference Cystic Fibrosis Center, Meyer Children’s Hospital, Viale Pieraccini 24, 50139 Florence, Italy; vito.terlizzi@meyer.it (V.T.); maria.cavicchi@meyer.it (M.C.C.); annasilvia.neri@meyer.it (A.S.N.); valeria.galici@meyer.it (V.G.); gianfranco.mergni@meyer.it (G.M.); 2Tuscany Support Cystic Fibrosis Service, Department of Pediatric, Leghorn Hospital, Viale Vittorio Alfieri 36, 57124 Leghorn, Italy; matteo.botti@uslnordovest.toscana.it; 3Diagnostic Genetics Unit, Careggi University Hospital, Largo Piero Palagi 1, 50139 Florence, Italy; centronec@aou-careggi.toscana.it; 4Department of Pediatric, Santa Chiara Hospital, University of Pisa, Via Roma 67, 56126 Pisa, Italy; diego.peroni@unipi.it; 5Department of Pediatric, Meyer Children’s Hospital, University of Florence, Viale Pieraccini 24, 50139 Florence, Italy; filippo.festini@unifi.it

**Keywords:** cystic fibrosis, salt-loss syndrome, metabolic alkalosis, false-negative cystic fibrosis screening, neonatal screening, diagnosis cystic fibrosis, *CFTR* mutation screening

## Abstract

Cystic fibrosis (CF) is a life-threatening and common genetic disorder. Cystic fibrosis newborn screening (CF NBS) has been implemented in many countries over the last 30 years, becoming a widely accepted public health strategy in economically developed countries. False-negative (FN) cases can occur after CF NBS, with the number depending on the method. We evaluated the delayed diagnosis of CF, identifying the patients who had false-negative CF NBS results over 26 years (1992–2018) in Tuscany, Italy. The introduction of DNA analysis to the newborn screening protocol improved the sensitivity of the test and reduced the FNs. Our experience showed that, overall, at least 8.7% of cases of CF received FNs (18 cases) and were diagnosed later, with an average age of 6.6 years (range: 4 months to 22 years). Respiratory symptoms and salt-loss syndrome (metabolic hypochloremic alkalosis) are suggestive symptoms of CF and were commons events in FN patients. In Tuscany, a region with a high *CFTR* allelic heterogeneity, the salt-loss syndrome was a common event in FNs. Therefore, we provided evidence to support the claim that the FN patients had *CFTR* mutations rarer compared with the true-positive cases. We underline the importance of vigilance toward clinical manifestations suggestive of CF on the part of the primary care providers and hospital physicians in a region with an efficient newborn screening program.

## 1. Introduction

Cystic fibrosis (CF) is a life-limiting autosomal recessive disorder that affects about 1 in 2500–4000 babies in the Caucasian population [1]. The classical CF phenotype is characterized by lung disease (bronchiectasis with persistent airway-based infection and inflammation) and exocrine pancreatic insufficiency associated with nutrient malabsorption, contributing to impaired growth and undernutrition [2]. CF is due to mutations in the gene encoding for the cystic fibrosis transmembrane conductance regulator *(CFTR*) anion channel [3]. The most common *CFTR* mutation associated with cystic fibrosis is Phe508del (c.1521_1523delCTT, DF508) [4]. Although the pancreatic status is closely related to the *CFTR* genotype [5], there is a wide clinical heterogeneity in CF patients, even between those carrying the same *CFTR* genotype or between siblings with CF [6].

The survival of individuals with CF has increased over the last 30 years, with a current predicted median age of survival of 43 years [7]. This is due to many factors, including earlier diagnosis through cystic fibrosis newborn screening (CF NBS) [8]. NBS strategies have been implemented since the early 1970s, and CF NBS is currently a widely accepted public health strategy in economically developed countries. All CF NBS strategies that are currently performed in NBS programs start with immunoreactive trypsinogen (IRT) on blood spots, usually in combination with a second IRT test (IRT2), pancreatitis-associated protein (PAP) on blood, or *CFTR* mutation analysis (DNA) [8]. A pancreatic meconium lactase assay (LACT) is an additional strategy in CF NBS programs that is used in some regions after the first immunoreactive trypsinogen (IRT1) dosage [9]. The European Cystic Fibrosis Society established that NBS programs should aim for a minimum sensitivity of 95% [10]. Individuals identified by NBS can be diagnosed with CF by a sweat chloride value ≥60 mmol/L and/or by the identification of two CF-causing mutations in trans [11]. Because CF NBS is widespread, most CF diagnoses presently occur in asymptomatic or minimally symptomatic babies following a positive NBS result [12,13,14], and this unequivocally benefits short- and long-term outcomes [15]. The presence of NBS may lead to an erroneous reassurance that all CF cases are identified at birth [16]. Maclean et al. showed that access to CF NBS seems not to result in a delay in diagnosis or poorer outcomes in those children for whom CF was not detected by CF NBS [17], but the question is debated [18,19]. Estimating the annual number of false negatives (FNs) in the CF NBS program is a complex challenge that depends on the sensitivity of the NBS method. It is essential for NBS laboratories and CF referral centers to have feedback on new CF cases not diagnosed by screening that allows the quality of the NBS program to be evaluated. Fritz et al. estimated that there were 70 FNs in 4,038,560 screened newborns per year in the USA using either the IRT/DNA or IRT/IRT strategy [19]. Rock et al. described the many possible causes of FN results that can occur after newborn screening for CF [20]. Farrel et al. declared that every physician’s first duty is to diagnose CF accurately and promptly because the “diagnosis is the first step of treatment” [21]. To limit irreversible organ pathology, a timely diagnosis of CF for a precocious institution of CF therapies can greatly benefit these FN patients. Several case reports of delayed diagnosis after a false-negative CF NBS have been described in the current literature [20,22,23,24]. The study of Padoan et al. analyzed 15 FN patients of the same CF center in Lombardy, a region in the northwest of Italy [25]. Calvin et al. reviewed 30 years of CF NBS in East Anglia, England, and evaluated 29 FN patients, including 10 FN patients with meconium ileus [26].

In Tuscany, a region in central Italy consisting of 3.7 million inhabitants, CF NBS has been carried out from 1984 in the CF Center of Florence [27,28]. In our center, all patients, who were residents of Tuscany, were also referred for the sweat test if they had symptoms suggestive of CF. We retrospectively evaluated CF patients who were born in Tuscany in the 1992–2018 period and who had negative results in CF NBS but who were later diagnosed because they had symptoms suggestive of CF. The aim of the study was to investigate FNs through the following characteristics: the symptoms and signs suggesting CF, the age of diagnosis, the current pancreatic status, the genotypical features, and the results of sweat tests.

## 2. Materials and Methods

From 1992 to 2018, we retrospectively reviewed all the results from the CF NBS database of the Regional Referral CF Center in Florence. The previous data were not analyzed in this study because the CF NBS protocol before 1992 was performed using a test with low sensitivity (<85%). Since 1992, the NBS has been performed by an immunoreactive trypsin (IRT) assay on dried blood spots, which is followed, if positive, by IRT retesting (IRT2) and meconium lactase dosage (protocol IRT1/IRT2 + LACT).

The cutoff of IRT1 was set at the 99th percentile, changing every 4–6 months according to NBS results and on the basis of seasons, as described in literature [29,30]. In 1992–1993, we used a colorimetric method for dried blood spot analysis of IRT1 (Medical System, IRT1 99th: 55 ng/mL). Successively, fluorimetric methods were used for IRT1 analysis (Delfia in 1994–2005, IRT1 99th percentile range: 54–62 ng/mL; Auto-Delfia in 2006–2013, IRT1 99th percentile range: 57–63 ng/mL). Since 2014, the IRT1 analysis has been performed by using the GSP instrument (Perkin Elmer), with an IRT1 99th percentile range of 49–50 ng/mL. Meconium lactase was detected by glucose production after incubation of meconium with lactose (cutoff 0.5 U/g), as described from Pederzini et al. [9]. The IRT2, taken at 3 or 4 weeks of age, followed the same procedure as the initial IRT1 assay (1992–2013 cutoff 29 ng/mL; 2014–2018 cutoff 23 ng/mL).

Since 2011, the NBS algorithm has been integrated with DNA analysis (protocol IRT1/IRT2 + LACT + IRT1/DNA); the overall Tuscany CF NBS strategy is illustrated in Figure 1 [31]. The detection rate of *CFTR* panel mutations through NBS was 80% until 2015 (38 CF-causing mutations were screened), 88% from 2016 to May 2018 (66 CF-causing mutations were screened), and 90% from May to December 2018 (272 CF-causing mutations were screened). In the database, we evaluated the total number of newborns screened, the true positives, the false positives, the true negatives, and the CF newborns with meconium ileus. In true-positive cases, all the patients with cystic fibrosis screening positive inconclusive diagnoses (CFSPIDs) who had developed CF during the follow-up (until 31 December 2019) were included [31,32].

Subsequently, with an extensive review of our clinical chart, we isolated the CF patients born in Tuscany over these 26 years but with negative screening (FNs) and a CF diagnosis preceding 31 December 2019. CF was first clinically suspected, following which a sweat test was performed according to the consensus statement [11]. The sweat chloride levels were tested according to the guidelines provided by the classic Gibson and Cook method [27] in the single laboratory of the Florence CF Center. The signs and symptoms of FN patients that prompted the physician or pediatrician (in primary care or in hospital care) to suspect CF were extensively examined from clinical charts. We also retrospectively assessed the age at diagnosis (in years) of the FN patients and the values of sweat chloride. The clinical symptoms revealed in our databases have been divided into these 10 categories:Respiratory (bronchopulmonary infections, chronic cough, expectoration)Gastroenteric (irregular defecation, frequent episodes of diarrhea)Failure to thrive (not meeting standards of growth)Salt-loss syndrome (dehydration with metabolic hypochloremic alkalosis)Nasal polyposisPancreatitisLiver diseaseDiabetesDistal bowel obstruction syndrome (DIOS)Azoospermia

Pancreatic sufficiency (PS) was defined based on at least two values of fecal pancreatic elastase being higher than 200 μg/g, measured independently of acute gastrointestinal diseases [33].

A genotype assessment was performed in all CF patients. We carried out gene sequencing (98% detection rate) in all CF patients, where one mutation was found after a first-level analysis of NBS. In all cases, an informed consent form was signed.

We analyzed the *CFTR* mutation alleles of FN patients and compared these with the *CFTR* alleles of true-positive patients. We compared the results with statistical analysis (chi-squared analysis with Yates correction) of the prevalence of the same mutations, which were more frequent in Tuscany for the true-positive cases than the FN cases. A *p*-value of less than 0.05 was considered statistically significant.

## 3. Results

In Tuscany, 746,686 newborns were screened in the 1992–2018 period. Throughout the 26-year period, both NBS protocols and diagnostic devices changed over time. In 2011, the IRT1/IRT2 + LACT algorithm was integrated with the analysis of mutations of the *CFTR* gene (IRT1/DNA), as reported in Figure 1. The sensitivity changed from 87.50% (1992–2010) to 96.15% (2011–2018). The CF incidence in the region over these 26 years was 1:3696 (1:3600 in the 1992–2010 period and 1:3939 in the 2011–2018 period). This showed that 202 patients had CF, of whom 162 (80.2%) were true-positive cases, 18 (8.9%) were FN cases, and 22 (10.9%) had meconium ileus. The CF newborns with meconium ileus had positive CF NBS results in 16/22 cases and FN results in 6/22. Patients with meconium ileus were excluded from the FN group and considered separately. The complete results of CF NBS are reported in Table 1. The average age at diagnosis of 18 FN patients was 6.6 years (range: 4 months to 22 years; median: 0.9 years). The symptoms at diagnosis were respiratory infections (8/18, 44.4%), salt-loss syndrome (7/18, 38.8%), failure to thrive (6/18, 33.3%), diarrhea (4/18, 22.2%), and nasal polyposis (3/18, 16.6%). Data on the symptoms leading to CF diagnosis, IRT values, pancreatic status, sweat chloride values, and genotypes are reported in Table 2. All FN patients had an IRT1 below the 99th percentile cutoff. Of the true-positive cases, 73.4% (119/160) had pancreatic deficiency. The frequencies of *CFTR* mutations detected in Tuscany are reported in Table 3. The five most frequent *CFTR* mutations in Tuscany (DF508, G542X, L1065P, N1303K, and E585X) were significantly less common in FN cases (41.7% of alleles) than in true-positive cases (60.4% of alleles; *p* = 0.029). In the group of FN patients, there was also a significantly greater prevalence of alleles with unknown mutations (8.3% vs. 1.1%, *p* = 0.001).

## 4. Discussion

CF NBS offers the opportunity for early medical and nutritional intervention, leading to improved outcomes [8]. In Tuscany, two NBS strategies were employed during the 26 years of this study. In the period during which DNA analysis was employed in the NBS algorithm (2011–2018), fewer FNs occurred, and the sensitivity improved from 87.50% to 96.15%. Overall, the screening program has proved effective in detecting 90.0% of CF cases at birth. The occurrence of FN results in CF newborns with meconium ileus (6/22 in our results) is reported in the literature [34]. However, it is not a concern, as these patients must always undergo a sweat test [10]. The situation of FN patients without meconium ileus who manifest CF symptoms at an older age is completely different, as a delay in diagnosis can compromise the nutritional status and lung function in such cases [35,36].

A disadvantage of CF NBS, particularly in regions with high allelic *CFTR* heterogeneity, is the detection of false-positive cases and/or carriers of a single CF-causing mutation [8]. Moreover, with the introduction of DNA analysis to the NBS algorithm, the number of CFSPID cases has increased [37]. These aspects of NBS may cause diagnostic dilemmas and, consequently, unnecessary anxiety for the family [38]. The problems associated with identifying CFSPID and/or mild cases of CF on NBS are well discussed by Munck et al. [32].

Our results, similar to those of Calvin et al. [26], have shown that the cutoff of IRT1 is the main determinant of sensitivity.

Our study focused on the symptoms suggestive of CF in 18 patients born in Tuscany with FN results in CF NBS. The symptoms suggestive of a CF diagnosis that prompt physicians to prescribe a sweat test are various in the literature; malnutrition and/or respiratory symptoms were the most common at the time of diagnosis [20,25,26]. In case reports, the clinical symptoms of malnutrition and/or respiratory symptoms were also most common at the time of diagnosis [23,24]. In Tuscany, respiratory symptoms (cough and respiratory infections), often in association with other clinical features (e.g., gastroenteric symptoms, failure to thrive, salt-loss syndrome), were the most common (44.2%). Metabolic alkalosis was noted in the prescreening literature as a prominent symptom before CF testing was available [39,40]. Our results illustrate that, in our region, metabolic alkalosis was a frequent clinical event that led to CF diagnosis (38.8%) in FN cases. There could be an association between alkalosis and rare *CFTR* mutations such as S737F, which has already been shown to be associated with salt-loss syndrome at diagnosis [41]. These data are different from those in the study of Padoan et al., where only 3 out of 15 FN patients (20.0%) in Lombardy had metabolic alkalosis at diagnosis [25]. In the research of Calvin et al., carried out in a region of England, no cases with false-negative results had salt-loss syndrome at diagnosis [26]. Another explanation of the higher frequency of salt-loss syndrome in Tuscany could be that this region has a hotter climate in comparison to the regions of Northern Italy and England. In our study, during the period in which DNA analysis was included in the protocol, all FN patients had salt-loss syndrome at diagnosis. The age at diagnosis for FN patients ranged from 4 months to 22 years, and CF was always diagnosed within the first year of life in the case of salt-loss syndrome.

We can speculate that if these FN patients had been somehow diagnosed by NBS, thanks to early saline supplementation, they probably would not have had salt-loss syndrome, a life-threatening event for an infant with CF. After the first year of life, respiratory symptoms, nasal polyposis, and the failure to thrive, associated with gastroenterological symptoms such as diarrhea, should lead the pediatrician to suspect CF. Only three FN patients presented with classical CF disease at diagnosis: respiratory and gastroenteric symptoms associated with failure to thrive (I, Q, R patients in Table 2). Pancreatitis, liver disease, diabetes, and DIOS were not found to be onset CF symptoms for FN patients. Conversely, of the 15 FN cases in Lombardy [24], 2 had recurrent abdominal pain and acute pancreatitis-like symptoms at diagnosis. In the literature, only one FN case was described with liver disease (cholestasis) at diagnosis [22]. The presence of mild CF in FN patients must be reported, such as case A, in which the patient had no symptoms (the older brother with mild CF was born in the prescreening era), and case E, in which the patient was diagnosed at age 15 for mild respiratory symptoms. Usually, FNs have pancreatic sufficiency (13/18, 72.2%), which could reflect the less severe clinical features in these subjects compared to CF true-positive cases (pancreas sufficiency in 41/160, 26.6%).

When CF is suspected, the gold standard of diagnostic tests still remains the sweat test, which was frequently already abnormal upon the first determination in our series. Indeed, only three patients (C, O, P patients in Table 2) had a borderline sweat test upon first determination and were CF-positive (chloride ≥60 mEq) upon subsequent determinations. These data confirm the need to repeat sweat chloride testing over time, especially for patients carrying *CFTR* mutations with varying clinical consequences [42]. The presence of FNs, despite the presence of well-organized CF NBS, must recommend that one always keep a vigilant eye on the possibility that a patient may have CF.

The *CFTR* molecular analysis of true-positive cases and FNs revealed a high heterogeneity of *CFTR* allele mutations in Tuscany. DF508 accounted for only 45.6% of the CF cases, according to the data on the prevalence of about 50% of the alleles in the Italian population [43]. In the FN cases, the most common mutations in Tuscany (DF508, G542X, L1065P, N1303K, and E585X), were found more rarely. This is the reason it is essential, in regions such as Tuscany with a high *CFTR* allelic heterogeneity, to always adopt NBS that analyzes a large panel of mutations to keep the sensitivity at high levels. The difficulty in planning the NBS strategy lies in balancing greater sensitivity with specificity and positive predictive value, thus avoiding the detection of CF carriers and inconclusive diagnoses. A key aspect in regions such as Tuscany with high *CFTR* allelic variability is that all mutations detected by screening must be CF-causing mutations. Another consideration based on these data is that when the suspicion of CF is strong, it is always necessary to perform a sweat test first. It is not prudent to trust the genetic test exclusively, given the high number of rare mutations, mutations with variable clinical consequences, and unknown mutations.

## 5. Conclusions

This study confirms the evidence that CF NBS is a valid public health strategy to detect the majority of CF cases at birth (90% in 26 years). While the sensitivity of CF NBS has been improved over the years, especially with the introduction of DNA analysis to the NBS algorithm, CF was not detected by screening in some babies because it was under the IRT1 cutoff. These patients (in total, 18 cases), born in Tuscany from 1992 to 2018, showed symptoms suggestive of CF during infancy, childhood, adolescence, or later. An interesting finding is that 38.8% of the FN patients manifested salt-loss syndrome (hypochloremic alkalosis) in the first year of life (associated or not with respiratory or nutritional/digestive symptoms). The median age at the diagnosis of Tuscan FN patients was 0.9 years. FN patients frequently have rare or unknown *CFTR* mutations compared with true-positive cases, and this confirms that a sweat test should be the primary test. Our results also show the importance of feedback of CF diagnoses in the CF screening centers. This study highlights the need for an improvement of CF NBS and the importance of maintaining vigilance toward clinical manifestations suggestive of CF. Having performed an efficient NBS test does not preclude a diagnosis of CF. A delay in diagnosis for FN patients can be shortened by a vigilant clinician.

## Figures and Tables

**Figure 1 diagnostics-10-00446-f001:**
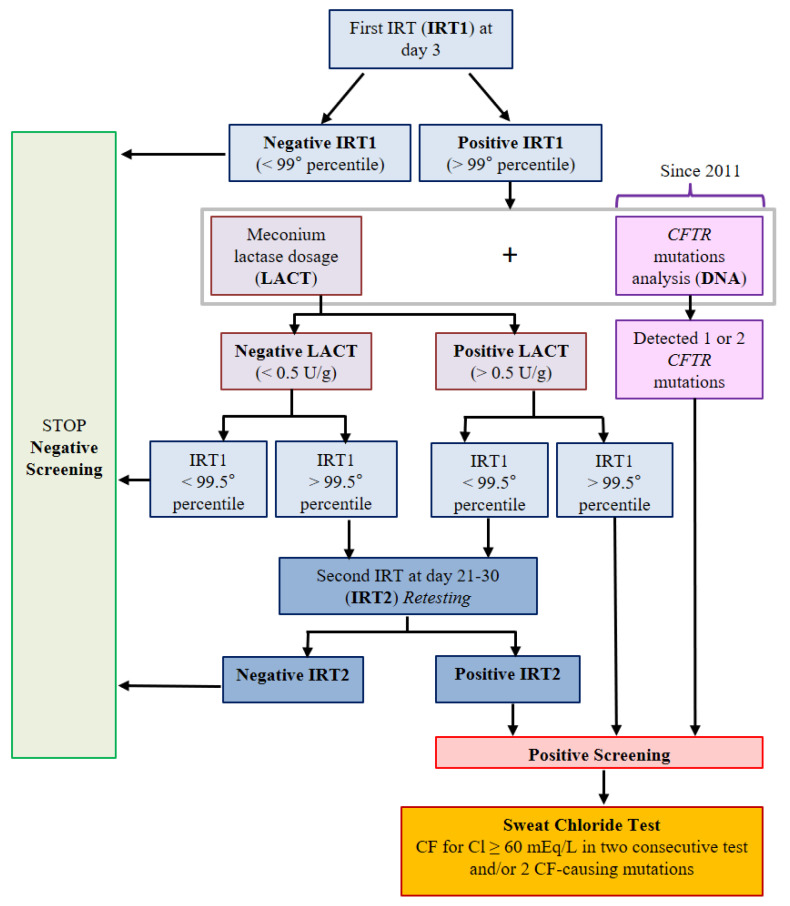
Cystic fibrosis newborn screening (CF NBS) algorithm adopted in Tuscany from 1992. The *CFTR* mutations analysis (on the right side of the graphs) was introduced in 2011.

**Table 1 diagnostics-10-00446-t001:** Data from the 1992–2018 Cystic fibrosis newborn screening (CF NBS). Percentages as related to all cystic fibrosis (CF) cases of column are included in parentheses.

	IRT1/IRT2 + LACT (1992–2010)	IRT1/IRT2 + LACT + IRT1/DNA (2011–2018)	Total
Newborns screened for CF	514,841	231,845	746,686
Newborns with negative screening	513,619	231,227	744,846
Non-CF newborns with negative screening (true negative)	513,603	231,225	744,828
CF newborns with negative screening (false negative)	16 (11.2%)	2 (3.4%)	18 (8.9%)
Number of sweat tests (newborns with positive screening)	1222	618	1840
CF newborns with positive screening (true positive)	112 (78.3%)	50 (84.7%)	162 (80.2%)
Non-CF newborns with positive screening (false positive)	1110	568	1678
CF newborns diagnosed with meconium ileus	15 (10.5%)	7 (11.9%)	22 (10.9%)
Sensitivity (%)	87.50	96.15	90.00
Specificity (%)	99.78	99.75	99.77
Positive predictive value (%)	9.16	8.09	8.80
Negative predictive value (%)	99.997	99.999	99.997

IRT: immunoreactive trypsin assay; LACT: meconium lactase dosage; DNA: *CFTR* panel analysis.

**Table 2 diagnostics-10-00446-t002:** Features of false-negative (FN) patients: genotype, symptoms, first immunoreactive trypsinogen dose (IRT1), age at diagnosis, pancreas status, and sweat test.

FN Patients	Year of Birth	IRT1 µg/L (Cutoff)	*CFTR* Mutation 1	*CFTR* Mutation 2	Symptoms and Signs Suggestive of CF	Age at Diagnosis	Pancreas Status	Sweat Test (Chloride mEq)
A	1993	<55 (55)	G542X	D110H	asymptomatic (family history)	22 years	PS	89-80
B	1995	<58 (58)	DF508	E831X	nasal polyposis	20 years	PS	104-96
C	1996	<58 (58)	DF508	unknown	respiratory	20 years	PS	58-42-72-83
D	1997	<60 (60)	G542X	unknown	respiratory + nasal polyposis	17 years	PS	62-67
E	1999	<58 (58)	DF508	5T/12TG	respiratory	15 years	PS	78-100
F	1999	<58 (58)	DF508	unknown	failure to thrive + gastroenteric	4 months	PD	110-115
G	2001	39 (54)	DF508	G542X	nasal polyposis	5 years	PD	80-94
H	2001	37 (54)	S737F	Del exon 22–24	salt-loss syndrome + respiratory	1 year	PS	82-89
I	2003	48 (54)	DF508	DF508	respiratory + failure to thrive + gastroenteric	4 months	PD	110-153
L	2003	45 (54)	W1282X	DF311	salt-loss syndrome	5 months	PS	60-63-85-93
M	2005	47 (62)	DF311	Del exon 2	salt-loss syndrome + failure to thrive	4 months	PS	75-71
N	2006	52 (57)	L206W	3659delC	salt-loss syndrome	10 months	PS	90-65-156
O	2006	41 (57)	L1065P	5T/13TG	respiratory	12 years	PS	52-61-40-69-101
P	2008	31 (60)	S737F	S737F	salt-loss syndrome	7 months	PS	45-46-93-64-98
Q	2010	51 (63)	DF508	DF508	respiratory + failure to thrive + gastroenteric	4 months	PD	102-96
R	2010	54 (63)	DF508	Delexon 22–24	respiratory + failure to thrive + gastroenteric	10 months	PD	125-125
S	2015	15 (50)	E831X	G126D	salt-loss syndrome	11 months	PS	67-74-70
T	2016	27 (49)	N1303K	L997F	salt-loss syndrome + failure to thrive	11 months	PS	60-58-84-67

PS: pancreas sufficiency; PD: pancreas deficiency.

**Table 3 diagnostics-10-00446-t003:** Frequencies of *CFTR* mutations in 360 cystic fibrosis *CFTR* alleles in Tuscany (324 *CFTR* alleles of true positive cases and 36 *CFTR* alleles of FNs).

True-Positive Cases		False-Negative Cases	
Mutations	Frequency (%)	Mutations	Frequency (%)
DF508	45.6	DF508	27.7
G542X	5.5	G542X	8.3
L1065P	3.7	S737F	8.3
N1303K	3.1	Delexon 22–24	5.5
E585X	2.5	L1065P	2.7
2789+5G->A	2.5	W1282X	2.7
Delexon 22–24	2.2	N1303K	2.7
R347P	2.2	Other rare mutations	33.3
W1282X	2.2	Unknown	8.3
T338I	2.2		
R553X	1.8		
S737F	1.3		
2183AA->G	1.3		
Other rare mutations	22.8		
Unknown	1.1		

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
