# Peer review of "Clinical and Genotypical Features of False-Negative Patients in 26 Years of Cystic Fibrosis Neonatal Screening in Tuscany, Italy"

_diagnostics, 2020, doi:10.3390/diagnostics10070446_

Round 1

Reviewer 1 Report

Summary

In this observational manuscript, the author’s report screening statistics for a region of Italy. The author’s find that false negative tests were less frequent after an adjustment in testing protocol in 2011. Alleles that were missed were likely rare alleles, although the Table should be formatted to make it apparent that the missed allele was not a common allele like dF508. The largest adjustment to the manuscript is the report that metabolic alkalosis was a surprise finding for undiagnosed CF as this was a prominent symptom in the early CF literature (PMID:13633369 and PMID:5132305; one of which is Dorothy Andersen’s seminal work) before CF testing was available.

Major Comments

Line 155-164. Metabolic alkalosis should not have been unexpected. Metabolic hypochloremic alkalosis has been noted in the pre-screening literature as a primary symptom of CF in pediatric case studies (for example Dorothy Andersen’s classical paper PMID:13633369 and PMID:5132305) please note this and other relevant studies. Please adjust.

Line 22-24. Are similar sentences. Do the authors mean “Respiratory symptoms and salt loss syndrome (metabolic hypochloremic alkalosis) are suggestive symptoms of CF and were commons events in FNs. Therefore, we provided…

Table 4. Is there a way to provide the undetected allele percentage as a 3rd subcolumn under False negative cases? The table looks like DF508 went undetected, even though it’s likely the 2nd allele that went undetected. 

Minor Concerns

The use of abbreviations FNs, TPs, etc. slow the flow of the manuscript. Spelling these terms out would be helpful to the reader. Alternatively, keep the widely used abbreviation “FNs” and spell out TPs, FPs, TNs.

Verify that a single convention for numbers are used. In most places no comma is used for thousands, however within table 1 a decimal point is used for 2 values. P values typically use a decimal point (.), but line 133 uses a comma (,). Figure 1 uses (,) for diagnostic numbers, but (.) is used in the manuscript. Table 4 uses (,).

Table 2 Chloride is misspelled

Table 1 “No CF” should read NonCF”, further the ordering is not intuitive. False negatives should be presented after true negatives, then # of sweat tests, followed by True Positive, followed by False Positive. The incidence of meconium ileus good to report, but its not clear how it relates to screening. If kept, then it maybe more useful to report as a percentage, or both an absolute and percentage.

Line 20 – protocol maybe a better word to use than algorithm also used in line 162

Line 66 – large is a relative claim just say “that analyzed 15 patients of FNs…”

Line 111 – delete space in “µ g/g”

Line 131-132 – more relative in Tuscany relative to where? Please explicitly state.

Line 123 should read “ileus”

Line 169 – change FC to CF

Line 178 – Positivized is unconventional, change to increased, also chloride is misspelled

Line 183 – Not sure what is meant by the sentence “Primary care…” this can be omitted, but could be restated if the author’s think its of importance.

Line 190 – change FC to CF

Author Response

Dear Reviewer 1,

the suggestions offered by you have been helpful. We also appreciated your comments regarding the abstract and the salt loss syndrome. We apologize for lexical errors.

We have the manuscript revised a second time by an English mother tongue teacher.

All changes requested by you and the other reviewers in the manuscript are in blue.

Major comments

Line 190-192: we have changed this sentence and inserted these references on metabolic alkalosis in pre-screening era. The new is that in the screening era, false negative subjects in our region had metabolic alkalosis. In other studies on false negatives (Padoan et al 2002, Calvin et al 2012) salt loss syndrome is not so common in FNs.

Line: 22-24: we have changed the sentence

Table 4: Table 4 now is “table 3” (the reviewer 3 recommended deleting table 3 because was too similar to table 2). In the current table 3 (ex-table 4) there are already the frequencies of CFTR mutations of true positive cases (1st and 2nd column) and FNs (3rd and 4th column). These mutations, in FNs, are detected always after screening, concomitant with CF diagnosis, with a genetic analysis.  When CFTR analysis revealed no mutations, mutations are considered unknown (in the lasts rows of Table 3). The genotype of every single FN case are reported in Table 2.

Minor concern

- We have changed the abbreviations: FNs remain, and spell out TPs, FPs and TNs

- We have corrected the points and commas in all manuscript

- We have corrected chloride in all manuscript

-  Thank you for your observation. To clarify, we added in Table 1, in every column, the percentages related to all CF cases of column

- Line 20 and 199: we have changed it to “protocol”

- We have deleted “large”

- Line 127: we have deleted the space

- We have changed FC with “CF”

- We have changed with “ileus”

- Line 219: we have changed it

- In the abstract 26-28: the “primary care” is the term to highlight that surveillance is necessary not only in the hospitals care but also in general practitioners.

Sincerely,

Giovanni Taccetti, Matteo Botti

Reviewer 2 Report

Overall this is a very useful paper for cystic fibrosis screening programmes. As mentioned, CF has been included in newborn screening programmes for many years. Especially after the discovery of the CFTR gene and protein the phenotype of CF has become much broader and most screening programmes don’t have a clear definition of their target condition (many wishing to detect as many cases as possible) although it is arguable that the detection of mild cases by newborn screening has any clinical benefit and may cause unnecessary anxiety (as does the labelling of infants with positive screens but without clear results in diagnostic testing with a ‘diagnosis’ of CFSPID or similar (Johnson et al 2019). It would be helpful if the authors could define the target condition for screening and discuss the harms of missing cases such as cases A-E in Table 2. Case A also raises the question of the appropriateness of relying on a screening test result when there are known risk factors for CF present eg family history (some public health screening purists would say screening should only be offered to people not at increased prior risk of having the screened disorder …….) and this could be addressed eg in the discussion. Also it would be interesting in Table 2 to have an indication of the newborn screening IRT (%-ile or MoM ideally as the numbers from different kits vary) to see how a drop in cutoff might affect the sensitivity. Or not.

The authors suggest (L186- ) that screening sensitivity can be improved by adding more mutations and this is correct but could be qualified by the need to balance increased sensitivity with reduced specificity and thus detection of more carriers of CF (a possible harm).

Minor specific issues are -

L16-17 false negative cases define test sensitivity so sentence as it stands is a tautology. Suggest reword eg ….. after CF NBS, the number depending on the method.

L50-52 my reading of this document (consenus guideline) suggests that a sweat test is essential to establish the diagnosis and use of mutation analysis without a sweat chloride value is suggested only to make a presumptive diagnosis so as not to delay treatment while a sweat test is awaited.

L83-85 needs word like disease or CF before causing – (eg 38 disease-causing mutations …)

L111 close up µ and g

L123 … of whom 162 … (which refers to animals and things, whom to people) or could reword sentence to be xxx had positive tests of which 162 were TP …..

L132 suggest replacing inferior with eg less common

L160 is the intent hotter and wetter? Hotter and warmer is repetitive.

L178 positivised isn’t a usual word and chloride is misspelled and unit incorrect Suggest ….. first determination which became positive (chloride …… mEq/L ) on subsequent ……

Author Response

Dear reviewer 2,

we appreciate the comments and advices dedicated to our manuscript. Thank you for pointing this out.

We have the manuscript revised a second time by an English mother tongue teacher.

All changes requested by you and the other reviewers in the manuscript are in blue.

In particular, we have added to the discussion the advice you have consider on the various aspects of screening, both positive and negative (line 175-180 and line 211-215).

The case A is a very particular case: this patient has an older brother, born in the Tuscany pre-screening era, whom we diagnosed with CF for the presence of azoospermia. Once CF was diagnosed in the brother, family members performed the sweat test and in this way case A arrived at the diagnosis at age 22 (positive sweat test, asymptomatic and NBS negative).

In our region, all first degree relatives of CF patients carry out the sweat test directly, regardless of the screening (which they do anyway).

From line 88 to 97 now we have reported in detail the various laboratory techniques that have followed and their cut off ranges. We added in Table 2 the cut-off of IRT of FNs.

Line 229-232. Your consideration is correct. In a region like ours, with extreme variability of the CFTR, we risk that by increasing the sensitivity too much, the specificity and positive predictive value will be reduced. It is a challenge that regions with less variability (as USA, UK, Canada…) in the CFTR gene do not have. It is essential to keep only CF-causing mutations in the screening mutation panel to avoid inconclusive diagnosis.

Minor comments:

We apologize for lexical errors

Line 16-17: We have changed it

Line 53: As suggested also by the reviewer 3 we changed the sentence adding “and/or”

Line 127: we have deleted the space

Line 141: we have changed with “whom”

Line 152: we have changed it with “less common”

Line 198: we changed it maintaining only “hotter”

Line 219: we changed it with “became positive”

Sincerely,

Giovanni Taccetti, Matteo Botti

Reviewer 3 Report

This work describes the results of the CF NBS program of Tuscany over 26 years, focusing on the analysis of false negative results. During this period (1992-2018) the program has analysed 746.686 newborns and detected 202 cases of CF: 162 true positives, 22 presented with meconium ileum and 18 were false negatives (8.7%). The results of this work describe the clinical symptoms and genotype of these false negative cases, drawing attention to the high frequency of salt loss syndrome (7/18) and the presence of less frequent CFTR pathogenic variants.

I consider that the publication of this work is valuable since it provides new information about the frequency of CF false negatives in neonatal screening programs and tries to predict the clinical and genetic factors that may distinguish these cases from true positives. However, it is surprising the lack of some data from the authors, which I consider relevant for the interpretation and understanding of these results and could help CF NBS programs to continue moving towards continuous improvement.

MAJOR COMMENTS

  1. Neonatal screening results (IRT1, LACT, IRT2) should be presented in the Results section, reported in Table 2 and discussed. I don´t know the reason why these data is not included. Perhaps, it would be easier to include them if the neonatal screening laboratory were invited to collaborate.
  2. Results. Describe the NBS results of the 22 CF cases that presented with meconium ileum (as they may present with false negative results). If these results are not available, indicate at least how many of them had a NBS false negative result.
  3. Results. Compare the prevalence of PS in CF false negatives which seems very high (13/18) with the prevalence of PS in true positives, which may be expected lower.

MINOR COMMENTS

-All the document. It is preferable to write the name of the gene CFTR in italics.

-All the document. It is preferable to use the term “pathogenic variant” or “disease causing variant” instead of “mutation”.

  1. Abstract. Include the absolute number of CF NBS false negatives detected (18), not only the percentage (8.7%).
  2. Abstract. Line 21. “…showed that, overall, 8.7% of cases of CF… “ change to “…showed that, overall, at least 8.7% of cases of CF…” Take into account that the follow-up of many NBS newborns with negative result is only one year and CF could be diagnosed later in time.
  3. Line 34. Change “The CF phenotype” to “The classical CF phenotype” as there are other milder phenotypes.
  4. Line 39. Change phe508del to Phe508del
  5. Line 46. “The most frequently performed laboratory test is immunoreactive trypsinogen…”. All CF NBS strategies that are currently performed in NBS programs start with the determination of IRT. Please, change this sentence.
  6. Line 47. “IRT……usually in combination with pancreatic meconium lactase assay (LACT). Reference 8 doestn´t support this CF NBS strategy (IRT + LACT). The authors say that IRT assay seemed to have better sensitivity than meconium albumin or lactase levels. Remove IRT + lactase assay as it is not a widespread used strategy. Include IRT/IRT strategy that is much more used.
  7. Line 51. Change “or” to “and/or” by the identification of two CFTR causing…”
  8. Line 51. Change “CTFR” to “CFTR”
  9. Lines 49-50. “…for a minimum positive predictive value of 0.3% and a minimun sensitivity of 95%”. Although I have checked that these are the most recent ECFS recommendations, all neonatal screening programs know that working with a PPV of only 0.3 is inadmissible, even more when CF is a common genetic disease. I personally consider this to be a mistake of the ECFS. Please, exclude the recommendation of PPV and mention only the related sensitivity.
  10. Line 59. Here, it could be mentioned the importance for NBS laboratories and CF referral centers having feedback on new CF cases diagnosed after screening, which allow evaluating the quality of the NBS program.
  11. Lines 66-67. The study of Padoan et al. Is not only the one that describes a large number of FNs. The East Anglian screening programme detected 29 false negatives over a 30 year period. Include reference: Calvin J. et al. Thirty years of screening for cystic fibrosis in East Anglia. Arch Dis Child. 2012.
  12. Methods. Include a brief description of the method used for the determination of IRT, LACT and DNA (eg. type of assay, commercial reagents or manufacturer, instrument).
  13. Methods. Between 1992 and 2011, were used the same percentile for IRT1 and same cutoffs for LACT (0,5 U/g) and IRT2 (29 ng/mL) than since 2011 to 2018? Please explain it.
  14. Methods. For IRT: I suposse you used a fixed cutoff that you updated periodically according to the recalculation of poblational percentiles. Please describe it.
  15. Line 87-88. Change “the true negatives (TNs) and the newborns with meconium ileus” to “the true negatives (TNs) and the CF newborns with meconium ileus”
  16. Results. Lines 128-129. As there are only 18 false negative cases, please present the results of presence of symptoms in proportions rather than in percentages (eg. respiratory infections (8/18) ).
  17. Line 133. Remove “p<0,05” as you are already reporting the p value (0,029).
  18. Figure 1. LACT units. Change “gr” to g”
  19. Figure 1. IRT2 units. Change “ng/ml” to “ng/mL”
  20. Figure 1. Change “Sweat Chloride Test” to “Sweat Chloride Test or 2 CFTR causing mutations”. This algorithm has been previously reported by you (reference 27).
  21. Table 2. Patient M. Change “Del esone 2” to “Del exon 2”
  22. Remove Table 3. I think that the information that provides is redundant.
  23. Legend of Table 4. Change “chromosomes” to “CFTR alleles”
  24. Line 187. Change “…were rarely found” to “were more rarely found”.
  25. Line 169, line 190 and Figure 1 (sweat test). Change “FC” to “CF”
  26. Line 199. Change “average age” to “median age” and the result “6.6 years” to “0.9 years”, because it highlights than half of the false negatives present symptoms and are diagnosed before the first year of life”
  27. Modify the conclusions according to the proposed changes.
  28. Include a section of “Author Contributions” at the back matter

Author Response

Dear reviewer 3,

thank you for the thoughtful comments and constructive suggestions, which help to improve the quality of this manuscript.

We have the manuscript revised a second time by an English mother tongue teacher.

All changes requested by you and the other reviewers in the manuscript are in blue.

  1. We discussed the results of screening of FNs. All our FN cases had IRT1 under 99.0th cut off. Unfortunately, before the year 2000, it is impossible for us to recover the exact values ​​of IRT1 of FNs, we have in our database only “under IRT1 cut-off”; after 2000 we reported in Table2 the exact value of IRT1 in FN cases. From line 88 to 97 we have reported in detail the various laboratory techniques that have followed over these 26 years and the cut off ranges.
  2. Regardless of newborn screening, all cases of meconium ileus perform a sweat test. The results of CF NBS in these patients now are reported in results (line 142-144). The discussion of this aspect of NBS is in line 171-174
  3. The percentage of pancreas insufficiency in true positive are reported now in line 149-150 and discussed in 211-215 line.

Minor comments

- CFTR changed, also in the Figure 1

  1. Abstract. We have included absolute number
  2. Abstract. Line 21 we have changed it
  3. We have changed it
  4. We have changed it
  5. We have changed it in line 47-49.
  6. We have corrected the sentence. The dosage of meconium lactase is common only in NBS strategies of same regions of southern Europe. Line 47-51
  7. We have changed it, line 53
  8. We have changed it in CF-causing mutation, line 53
  9. We have removed PPV of ECFS guidelines.
  10. We have added it in line 61-62
  11. Thank you for pointing out this important article. We have added it in the introduction (line 71-72) and in the discussion (195-199)

12. 13. 14. According to your advice we have expanded the parts of manuscript that describe the results of the screening and laboratory techniques. From line 88 to 96 now we have reported in detail the various laboratory techniques that have followed and the cut off ranges. We have adopted the changing of the IRT1 99th cut off according to the result of the previous screening (10,000 previous births, corresponding every 3-4 months) and according to the season as described by Farrel. The cut-off value of LACT remained the same cut-off. In Figure 1 now the CF NBS algorithm from 1992 to 2018 is shown, specifying in the caption that the right side of graphs, the CFTR analysis, was introduced since 2011. This being the only difference between the two periods, we preferred to maintain a unique figure of NBS algorithms.

  1. We have changed it in line 101-102
  2. We have change it in line 146-147
  3. We have removed it

18.19.20. We have changed all these in Figure 1

  1. We have changed it in table 2.
  2. We have removed the Table 3
  3. We have changed it in Table 4 (currently “table 3”)
  4. We have changed it in line 229
  5. We have changed it
  6. We have changed it
  7. We have changed the conclusions
  8. The role and detailed contribution of each author was already included at the moment of submission of the article to the Journal.

Sincerely,

Giovanni Taccetti, Matteo Botti

Round 2

Reviewer 3 Report

The authors have revised the article based on my suggestions.
For my part, it is ready for publication